# Portfolio Management with Reinforcement Learning

**Zikai Sun**
Department of Electronic Engineering
The Chinese University of Hong Kong
Shatin, Hong Kong
zksun@link.cuhk.edu.hk

**Yuting An**
Department of Mechanical and Automation
The Chinese University of Hong Kong
Shatin, Hong Kong
ytan@mae.cuhk.edu.hk

## Abstract

Portfolio management is a crucial trading task for investment companies in the market. In this work, reinforcement learning (RL) incorporating the transformer structure is combined with deep learning (DL) to build an automated portfolio management model. The proposed method uses the Sharpe ratio along with transaction cost as the reward and build an environment that contains the whole A-share market to train the RL agent. The result demonstrates that the trained strategy outperforms The Shanghai Composite Index and tradition baselines.

Video link:IERG5350 project video link

## 1   Introduction

Modern portfolio theory suggests that one should pay more attention to return per unit risk. People can reduce exposure to individual asset risk by simply holding a diversified portfolio of assets, which is called portfolio management. With the rapid development of big data techniques and quantitative trading, automated portfolio management strategy has become more crucial to investment companies.

The portfolio management problem is a sequential decision-making process over time series that allocates financial assets like stocks, bonds, options, cash, or more in the market. This process can naturally be solved by the reinforcement learning (RL) framework. And we aim to design a reinforcement-learning-based trading strategy which could achieve better performance than widely used baselines. The proposed method utilizes Deep Deterministic Policy Gradient (DDPG) as the framework of RL to learn the dynamics in the market and replaces the four Multi-layer Perceptions (MLP) in DDPG with Gated Transformer-XL (GTrXL) to well process the time-series stock data. The goal of the proposed method is to develop a RL-based automated portfolio management strategy that makes more profit while maintaining the risk in consideration of transaction cost.

## 2   Related work

Portfolio management could be treated as a Markov Decision Process (MDP). And, in recent years, reinforcement learning has been widely applied in solving this MDP problem, helping design advanced strategies through the interaction with the historic market data like price, volume, financial statements, and news[8].

Reinforcement learning approaches could be classified into three categories: critic-only, actor-only, and actor-critic approaches[1]. The critic-only approach estimates the state-action value, Q, according to the expected return when the agent takes different actions. Deep Q-Network (DQN) is a classical method[3], which approximates the state-action value with neural networks and extends the method to deal with continuous state space. The actor-only approach learns the policy (a mapping from state to action) directly instead of computing state-action value. So, it could work well

when the action space is continuous. Actor-critic is the most commonly used method in recent years. The Actor-network learns the policy and the critic network approximates the Q value simultaneously, which combines the advantages of these two methods. Deep Deterministic Policy Gradient (DDPG) is a popular framework that incorporates actor and critic network in serial. Xiong designs an algorithm based on DDPG and defeats the traditional min-variance portfolio allocation method and Dow Jones Industrial Average[8]. Kim develops an algorithm that combines the DDPG with the GTrXL for the portfolio optimization in US stock market[5].

Apart from the reinforcement learning approach, other elements like state, action, and reward function should be finely designed based on various data and market factors. For state design, asset close price and shares owned, along with the available cash during a certain period is neccesary[6]. Also, some technical analysis indicators like Moving Average Convergence Divergence (MACD), Relative Strength Index (RSI) are incorporated[2, 10]. Moreover, Ye predicted the price movement using Long-short Time Memory (LSTM) and handled the related news with Natural Language Process (NLP)[9]. The encoded results are then synthesized into the state with a simple asset price. For action design, Yang[2] took selling, buying, and holding of each stock as an action space. Besides, the net value proportion of assets in each time step is adpoted[9]. For reward function design, change of the portfolio net value is a good choice[8] and transaction costs are considered by Ye and Jiang.[4, 9] Sharpe ratio, which represents the relationship between return and risk is also an ideal objective function.

## 3 Problem Formation

In this section, we introduce how the portfolio management problem is modeled in the RL framework. And the state, action, and reward function are also defined in this section.

### 3.1 State

The minute-level stock price and the transaction volume of each equity in the China A-share stock market are chosen to construct the environment for the RL agent to learn. Three elements of the state are then considered: the current position, the market information, and the overall macroeconomic information.

(1) The current position includes the information of the prices of stocks $v_{i,t}$, the weight of holdings of stocks $w_{i,t}$. This is denoted as $s^* = [v, w]$

(2) The market information includes historical minute-level trading information of assets like price variation and trading volume. Several widely used technical indicators are also considered in this approach. Because most of the market participants utilize these indicators while trading, the market will definitely contain the pattern related to these indicators which could be extracted. The $i$ th stock's observation at time $t$ can be expressed as $x_{i,t} = \{open, close, high, low, volume, MACD, MA, ...\}$. Then for the RL system that takes $N$ steps, the input can be written as $X_{i,t} = \{x_{i,t}, x_{i,t-1}, ..., x_{i,t-N+1}\}$ For this part, we consider using a gate recurrent unit (GRU) to extract it into a feature vector. The learned model structure is denoted as $f_{w_1}$.

(3) The third element is some macroeconomic information like risk-free rate and the unemployment rate, and corporate financial/fundamentals data like PE and PB, since they are strongly related to financial market and are crucial in predicting economic crisis and prosperity. All of the data in this part could be expressed as $h_{i,t}$. Because of the limitation of time, they are not included in this research.

Finally, the system concats all encoded features together, and the synthesized state is then defined as

$$s_t = (s^*_{i,t}, f_{w_1}(X_{i,t}), h_{i,t})$$

### 3.2 Action

Unlike many algorithms that use buying, holding, and selling a certain asset as actions, it is more efficient and effective to define the action space as the net value proportion of assets that will hold for the next time step. The action denoted as $a_t = \{a_{0,t}, ..., a_{n,t}\}$, which means the holding weight

of the $i$th asset is re-allocated to $a_{i,t}$. At the same time, we limit the allocation ratio of every asset as

$$a_{i,t} \in [-k, k], \quad \sum_i^n |a_{i,t}| <= k$$

Where $k$ is the maximum leverage level, and many brokerage can provide 2 times leverage. By this definition, we can easily get each asset's net value proportion $w_{i,t+1}$ at the next time step, considering the stock price movement ratio $y_{i,t} = v_{i,t+1}/v_{i,t}$.

$$w_{i,t+1} = \frac{y_{i,t+1} \cdot a_{i,t+1}}{\sum_k y_{i,t+1} \cdot a_{i,t+1}}$$

## 3.3 Reward

On one hand, to achieve a balance between return and risk, the proposed system directly uses the Sharpe ratio as part of the reward function. Since the T-bill interest rate is nearly 0 in 2020, the risk-free rate of return could be ignored, the formulation odf the Sharpe ratio could then simplified as:

$$Sharpe\ ratio = \frac{E(R_p)}{Std(R_p)} = \frac{E(R_p)}{\sqrt{E(R_p^2) - (E(R_p))^2}}$$

where we use the log return to make it satisfy additivity.

$$E(R_p) = \frac{1}{T} \sum_{t=1}^{T} R_{p,t}, \quad R_{p,t} = \ln(\sum_{i=1}^{n} a_{i,t} y_{i,t})$$

On the other hand, considering that there will be trading losses or transaction costs, it is hoped that the system could reduce the number of unnecessary tradings as much as possible. The transaction cost is then defined as $c = \sum_{i=1}^{n} |a_{i,t} - w_{i,t}|$. With the learned strategy $\mu_\theta$, the system aims to maximize the objective function, formally written as:

$$\mu_{\theta_*} = \underset{\mu_\theta}{\arg\max}\ J_T(\mu_\theta)$$

$$s.t. \sum_{i=1}^{n} |a_{i,t} - w_{i,t}| < \tau$$

when the lagrange parameter $\beta$ is introduced, this can be written as

$$\mu_{\theta_*} = \underset{\mu_\theta}{\arg\max}(J_T(\mu_\theta) - \beta \sum_{i=1}^{n} |a_{i,t} - w_{i,t}|)$$

So, considering both risk-reward ratio and transaction cost, the reward function is then defined as follows:

$$r = \frac{\sum_{t=1}^{T} R_{p,t}}{\sqrt{T \sum_{t=1}^{T} R_{p,t}^2 - (\sum_{t=1}^{T} R_{p,t})^2}} - \beta \sum_{i=1}^{n} |a_{i,t} - w_{i,t}|$$

## 4 Methods and Algorithms

Since the portfolio management problem is with continuous action space, partial observability, and high dimensionality, the DDPG with 2D Relative-attention Gated Transformer is proposed and could be applied to A-share automated stock trading. The developed RL framework in Figure 1 includes the following elements: Actor-Critic Architecture, Gated Transformer-XL, Gradient Update Algorithm, Replay Buffer.

**Actor-Critic Architecture**   The framework essentially follows the structure of the DDPG to deal with the continuous action space. In detail, the architecture incorporates two actor and two critic neural networks. Both of them includes a behavior network that interacts with the environment and a target network that helps update the behavior network. The observed state $s_i$ is the input of the actor behavior network. The output action $a_i$, along with the observed state $s_i$, serve as the input of the critic behavior network and the network output is the $Q$ value. The structure of actor/critic target network is similar with the behavior one and the output $Q'$ is utilized to calculate the TD-error with $Q$ and $r_i$

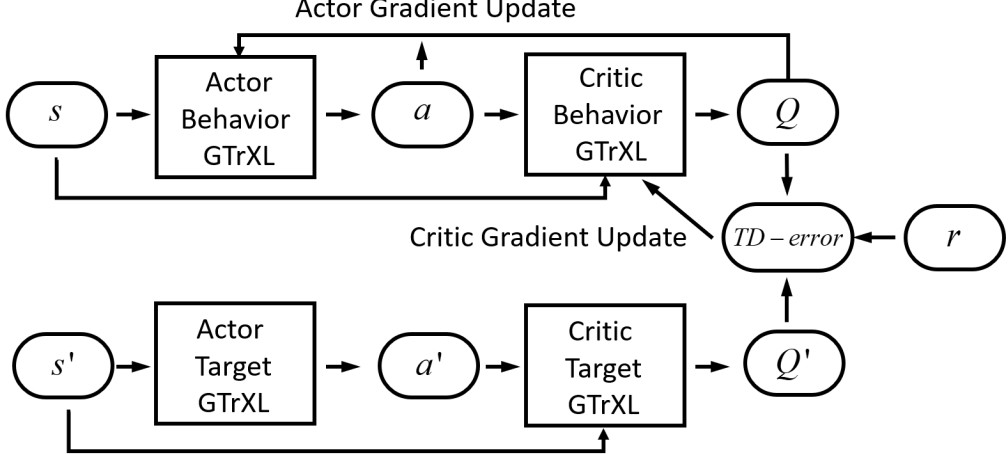

Figure 1: The RL framework of DDPG with GTrXL.

**Gated Transformer-XL**   In this work, Transformer encoders that have a robust structure to long term dependency of partial observability is utilized since many experiments have proved that self-attention architectures can deal better with longer sequences than recurrent neural networks(RNN) while avoiding the gradient vanishing or exploding during the optimization process[7]. Specifically, a variation of Transformer called 2D Relative-attentional Gated Transformer (RG-Transformer) is used as a core part of the behavior/target actor and behavior/target critic for high dimensional portfolio data[5].

The final GTrXL layer block is written below:

$$\overline{Y}^{(l)} \& = \text{RMHA}(\text{LayerNorm}([\text{StopGrad}(M^{(l-1)}), E^{(l-1)}]))$$

$$Y^{(l)} \& = g_{\text{MHA}}^{(l)}(E^{(l-1)}, \text{ReLU}(\overline{Y}^{(l)}))$$

$$\overline{E}^{(l)} \& = f^{(l)}(\text{LayerNorm}(Y^{(l)}))$$

$$E^{(l)} \& = g_{\text{MLP}}^{(l)}(Y^{(l)}, \text{ReLU}(\overline{E}^{(l)}))$$

where $g$ is a gating layer function.

**Gradient Update Algorithm**   The target return $G_i$ for the i-th sample from replay buffer is:

$$G_i = r_i + \gamma Q'\left(s_i', \mu'\left(s_i' \Big| \theta^{\mu'}\right) \Big| \theta^{Q'}\right)$$

where $s_i$, $a_i$, $r_i$, $s_i'$ and $\gamma$ are the state, action, reward, next state, and discount factor, respectively. The critic weights $\theta^Q$ is updated by minimizing the loss from temporal difference error between $G_i$ and $Q\left(s_i, a_i \big| \theta^Q\right)$:

$$L = \frac{1}{N} \sum_i \left(G_i - Q\left(s_i, a_i \big| \theta^Q\right)\right)^2$$

Also, the policy gradient to update the actor weights θμ are calculated using the chain rule as:

$$\nabla_{\theta^\mu} J \approx \frac{1}{N} \sum_i \nabla_{\theta^\mu} Q\left(s, \mu\left(s|\theta^\mu\right)|\theta^Q\right)|_{s=s_i, a=\mu(s_i)}$$

$$= \frac{1}{N} \sum_i \nabla_a Q\left(s, a|\theta^Q\right)|_{s=s_i, a=\mu(s_i)} \nabla_{\theta^\mu} \mu\left(s|\theta^\mu\right)|_{s=s_i}$$

Finally, the target actor weights $\theta^{\mu'}$ and target critic weights $\theta^{Q'}$ is updated slowly with $\tau\theta^\mu + (1-\tau)\theta^{\mu'}$ and $\tau\theta^Q + (1-\tau)\theta^{Q'}$, respectively, where $\tau$ is the target update rate.

**Dual Replay Buffers**    To accelerate the training, an asynchronous learning method with dual memories has been employed. The experience replay buffers consists of one memory that saves all the trajectories and h-memory that saves trajectories with high rewards.

# 5   Experiments

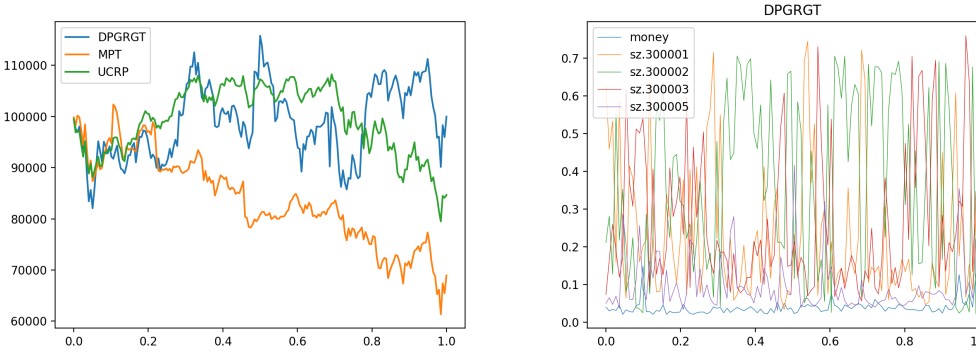

Figure 2: Left: returns of different agent. Right: weights when testing.

**Dataset preprocess**    We use the whole China A-share stock market as our stock pool, which contains 4066 stocks. The time span of our dataset is from 2010/01/02 to 2020/11/1. We consider this market because (1) over 87.78% of stocks are held by individual investors at 2020 Q2, which may have more emotional patterns and relatively easier to learn by the algorithm. (2) we can get the minute-level data for all stocks, which guarantee that we have enough data to train.

Data preprocessing has a great influence on the experimental results. For the daily data of each stock, we first obtain its basic minutes' data such as its open, high, low, and close price. We also consider the volume information. For the trading information and technical indicators, we divide data by the closing price of the previous day. We do this because we want to pay more attention to the trend of price rather than the price itself. Also, we calculate 20 different technical indicators on the scale of 5 minutes and 1 day separately, such as MA, EMA, MACD, etc. Macro-economy information and financial data are neglected and will be considered in future research.

**Experiment result**    We use all the stocks to train the agent. For each episode, the environment randomly selects 4 stocks as the dataset, and the model outputs the best weights of holding for each stock. When testing, we fixed 4 stocks as stock pool, and run the agent on the testing dataset.

We compare our results with some current baseline strategies (such as Modern Portfolio Theory (MPT) strategy, Equally Weighted Rebalance (EWR, called UCRP in figure label) strategy, etc). The comparison of the return and the stock weights (action) of each equity is demonstrated in Figure 2.

In future work, we plan to include more specific financial indicators like historical PE(TTM), PB and consider macro-economy information like total social financing of China and unemployment rate. More stocks will be utilized in testing and more experiments will be conducted.

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
