# OpenReview forum: "Portfolio Management with Reinforcement Learning"
_CUHK.edu.hk/2021/Course/IERG5350_

### Official Review · AnonReviewer2 · 2020-12-14
**Good paper with more analysis on the experiment results needed**

**Rating:** 8
**Confidence:** 5

**Review:**

Summary: This paper focuses on portfolio management with the help of RL. Overall, this paper is centered around the course content and well-written.

General:

Significance: The main contribution of this paper is to use DDPG and GTrXL to solve the tricky portfolio management problem. It makes contributions to this area since I think some research works are similar to the solution in the paper. But it is pretty fine since it meets the requirements of a course project.

Novelty: This paper obviously has some novelty in general since the combination of DDPG and GTrXL is totally new from my point of view. Also, the definition of reward and action is difficult to handle in this problem but the authors show some novelty in that as well.

Technical quality: The paper's technical quality is good. The reward is clearly defined. In addition, the definition of all the equations for each building block are well defined with clean formula.

Clarity: This paper is clearly written and somewhat well organized. The paper seems to be written in a rush as there is no conclusion section. Also, the analysis of the experiment section is not enough. It still confuses me after carefully reading it.

Specific:
a. Pros:
1. The model architecture is reasonable to me and has many original ideas.
2. All the equations are well defined.
3. Figure 1 is very illustrative and informative. However, it can be enlarged to make it clear.

b. Cons:
1. No GitHub codes are available.
2. More experiments are necessary.
3. More analysis of experiment results are necessary.
4. No conclusion.

Details:
The last line of section 3.1, 'concats' should be 'concatenates' in a more formal way.

---

### Official Review · AnonReviewer3 · 2020-12-15
**Well-defined task; Good idea and implementation; Insufficient analysis on experimental results; Hasty finish on paper writing.**

**Rating:** 8
**Confidence:** 4

**Review:**

In this project, the authors implement a reinforcement learning framework to address the problem of portfolio management task. The major contribution is that the authors incorporate the 2D Relative­-attention Gated Transformer­XL network to the DDPG framework as an encoder to better capture the long term dependency of partial observability of the stock information.

The idea is innovative to some extent, as the authors introduce the Gated TransformerXL network to replace the original MLP layers in DDPG algorithm considering the characteristic of the task. Although idea of incorporating structures like LSTM and Transformers to basic RL framework to deal with time-series data is proposed in some literature, the idea and implementation is good enough to meet the course project's requirements.

The paper is tidy, well laid out and consistently formatted. The problem is well modelled and illustrated, where the authors carefully choose the definition of states, action and reward. And the experiments with results also clearly shows that the proposed method has a good performance.

However, there are still some problems:
1) There could be more analysis on experimental results. And a clear illustration of results figure should be given so that the reviewers can better understand the performance of the proposed algorithm in comparison with baselines.
2) More efforts should be paid for the experiments and result analysis, as it is a little difficult for reviewer to understand the figure 2. In reviewer's idea, providing the meaning of horizontal and vertical axis of the result diagram can also help.
3) The authors could add an additional section to state the future work instead of placing them at the subsection of experiments.
4) It would be better to provide a conclusion section to help the reviewers quickly recap the idea and emphasise the contribution of the paper.

Overall, the idea and implementation satisfies the requirement of this course very well in reviewer's opinion, but the authors should pay more efforts on experiments and analysis on results. The paper has a strong start and content but seems to have a hasty finish.

---

### Official Review · AnonReviewer1 · 2020-12-19
**Good implementation; need more analysis and illustrations**

**Rating:** 7
**Confidence:** 3

**Review:**

Significance: This paper proposed a RL-based automated portfolio management strategy, which uses DDPG with GTrXL to process the time-series stock data. I think the research on this strategy is very important due to the current development of big data.

Novelty: This paper replaces the four Multi­layer Perceptions (MLP) in DDPG with Gated Transformer­XL (GTrXL). It’s novel to me. In addition, the authors define the action space as the net value proportion of assets that will hold for the next time step instead of buying, holding, and selling a certain asset.

Technical quality: Good. Clear defined formula and methodology.

Clarity: This paper is well-organized and clear to follow.

Specific:

a. Pros:
1. Use DDPG with GTrXL to process the time-series stock data
2. The description of the algorithm is very clear.

b. Cons:
1. It didn't analyze the result.
2. Need a Conclusion.